# Co-Training Method Based on Semi-Decoupling Features for MOOC Learner Behavior Prediction

Huanhuan Wang [1,2] , Libo Xu [1,2,*] , Zhenrui Huang [1,2] and Jiagong Wang [1,2]

[1] School of Computing and Data Engineering, NingboTech University, Ningbo 315000, China; whh@nit.zju.edu.cn (H.W.); huangzhenrui@nbt.edu.cn (Z.H.); 3190439064@nit.zju.edu.cn (J.W.)
[2] School of Computing and Data Engineering, Ningbo Institute of Technology, Zhejiang University, Ningbo 315000, China
[*] Correspondence: xlb@nit.zju.edu.cn; Tel.: +86-0574-88130956

**Abstract:** Facing the problem of massive unlabeled data and limited labeled samples, semi-supervised learning is favored, especially co-training. Standard co-training requires sufficiently redundant and conditionally independent dual views; however, in fact, few dual views exist that satisfy this condition. To solve this problem, we propose a co-training method based on semi-decoupling features, that is, semi-decoupling features based on a known single view and then constructing independent and redundant dual views: (1) take a small number of important features as shared features of the dual views according to the importance of the features; (2) separate the remaining features one by one or in small batches according to the correlation between the features to make "divergent" features of the dual views; (3) combine the shared features and the "divergent" features to construct dual views. In this paper, the experimental dataset was from the edX dataset jointly released by Harvard University and MIT; the evaluation metrics adopted *F*1, *Precision*, and *Recall*. The analysis methods included three experiments: multiple models, iterations, and hyperparameters. The experimental results show that the effect of this model on MOOC learner behavior prediction was better than the other models, and the best prediction result was obtained in iteration 2. These all verify the effectiveness and superiority of this algorithm and provide a scientific and feasible reference for the development of the future education industry.

**Keywords:** semi-supervised; co-training; semi-decoupling; feature importance; Pearson correlation coefficient





## 1. Introduction

The basic purpose of machine learning is to learn useful knowledge from data and to use inductive laws to analyze and predict future results. Machine learning methods are mainly divided into supervised learning, unsupervised learning, and semi-supervised learning [1,2]. Among them, supervised learning needs a large number of labeled data [3–6], while unsupervised learning does not need any prior knowledge, but it clusters similar samples into one category by fitting the internal distribution of unlabeled data [7–12]. With the rapid development of technology for data collection and storage, the acquisition of unlabeled data has become quite easy [3], but labeled data often require a lot of manual intervention and, in some cases, even professional advice. Thus, in most scenarios, the amount of labeled data are still insufficient for many practical applications. However, if only a small amount of labeled data are used, it is easy to cause an overfitting problem for the learning model and leads to a lack of good generalization ability. The challenge of using a large number of idle unlabeled samples to effectively enhance the generalization of the model has become one of the hotspots in the machine learning field [13,14].

Semi-supervised learning is an important research hotspot that combines supervised learning with unsupervised learning [15]. The aim is to use massive unlabeled data and limited labeled data to train the model, learn the potential information of the unlabeled

data, ensure the good performance of the model on the basis of supervised learning, and improve the generalization of the model [16,17]. According to different learning methods, it is mainly divided into four categories: a generative model-based method [18], a graph-based method [19], a semi-supervised support vector machine-based method [20], and a co-training algorithm [21,22].

Compared to other methods, co-training is simpler and more efficient. It must be assumed that there are sufficiently redundant and conditionally independent dual views in the problem domain. Here, "sufficiently redundant" means that each view contains enough information to produce an optimal model, and "conditionally independent" means that under the category label of given data, the two views are independent and do not interfere with each other. By using the dual view, two classifiers are trained from different perspectives, and then the data with the higher confidence are selected as a pseudo-label sample to add another view to achieve complementary advantages. They learn from each other and make progress together until they no longer change or reach a preset iteration.

Among the existing co-training studies, there are several models based on different learning algorithms, different data adoptions, different parameter settings, and different multi-view acquisitions such as tri-training and random segmentation. They all enhance the classification performance of co-training and reveal the intrinsic mechanism. However, the data obtained by these models' algorithms still suffer from conflicting feature attributes with poor relevance, difficulty in optimizing important features, and the limitations of irrelevant features in real application scenarios. These all directly affect the performance of classifiers and, so far, there has been no research to address these issues in depth. To solve this situation, we propose a co-training method based on semi-decoupling feature: segmentation of a single view based on feature attributes. The main steps are as follows:

(1) Calculate the feature importance of the dataset and rank them, and then select a number of the most important features as the shared features of the dual views;

(2) Calculate the correlation coefficients between features, process the remaining features according to the correlation coefficients, and then add them to the dual views as other features. While not reducing the data and ensuring the sufficiency of the data, separate the two views as independently as possible according to the feature correlation and difference;

(3) Combine shared features and "divergent" features to complete feature semi-decoupling, and then construct independent and redundant dual views for co-training.

This algorithm was tested on the edX dataset released by Harvard University and MIT, evaluated with *F*1, *Precision*, and *Recall*, and analyzed by multiple models, iterations, and hyperparameters. The experiments show that our model obtained better performance on MOOC learner behavior prediction. In addition, they also demonstrated the further effectiveness and superiority of the algorithm that is proposed and implemented in this paper.

In this paper, we aimed to disentangle the independent and redundant dual views on a known single view as much as possible to satisfy the assumptions of standard co-training multiple views, to avoid the problems of conflicting feature attributes and difficulties in optimizing important features, and to improve the performance of the classifier.

## 2. Related Work

Since 1998, when A. Blum and T. Mitchell first proposed the formal co-training algorithm, co-training has gradually become one of the most important mainstream directions in the field of semi-supervised learning [23]. The existing literature makes further improvements on the standard co-training. On the one hand, it improves co-training for single views and generates multiple sub-learners with differences by different learning algorithms [24], different data adoptions [25], and different parameter settings [26] so that multi-view learning can be accomplished even when the views are not redundant and the view features are not independent. On the other hand, there are also some studies around view segmentation to achieve co-training by differential multi-view acquisition [27].

The co-training model proposed by A. Blum and T. Mitchell first assumes that there are two fully redundant and conditionally independent views in the dataset. They use labeled samples to train different classifiers on these two views, and then select a number of samples with higher confidence from the unlabeled samples. After, those samples are added to each other's classifier to realize differences and complete an update. In this way, the whole process is continuously iterated until it reaches a certain stop condition. Subsequently, based on different learning algorithms, co-training, tri-training, co-forest, and co-trade variants have been derived [28–30]. References [31,32] suggested the tri-training algorithm; it neither requires fully redundant multiple views nor various types of base classifiers but combines semi-supervised learning and integrates learning mechanisms to obtain three labeled training sets by repeatedly sampling from labeled samples and generating three classifiers to achieve prediction on the same unlabeled sample. References [33,34] also proposed the co-forest algorithm, which uses a combination of several classification decision trees to ensure the difference and robustness of each classifier. Unlike the tri-training and co-forest models, the co-trade model follows the standard co-training premise assumptions [35,36]. It uses a data editing technique based on cutting-edge weight statistics to determine the labeling confidence. Then, the pseudo-labeled data with the lowest error rate is selected to join the next round of training data for another classifier. The algorithm provides the exact value of the error rate for each iteration and reduces the introduction of noisy samples during the training process.

There are also some studies around view segmentation, so as to achieve adequate and redundant multi-view acquisition [37,38]. The literature [39] has proposed a random subspace partitioning algorithm, which mainly discusses the influence of the numbers and dimensions of subspaces on the classification performance. The more random the subspaces, the worse the classification performance. The main reason for this is that the quality of the optimal subspace selection is limited by the irrelevant features. The literature [40] suggests a view adequacy-based segmentation algorithm, which is based on attribute simplification in rough set theory. In this algorithm, high-dimensional data are reduced to low-dimensional subspaces, and important attributes are added sequentially until the current mutual information and the original mutual information are equal. Reference [41] proposed a view independence-based segmentation algorithm. The core idea is to use the maxInd algorithm of graph, add mutual information index to measure the amount of information sharing between features, and realize multi-view acquisition. The literature [42] suggests an automatic segmentation algorithm, mainly by initializing the weights of two classifiers, and then the two classifiers split the single view based on the optimal loss function to obtain two new views.

### 3. Materials and Methods

*3.1. Co-Training*

In this section, we briefly review co-training and the relevant concepts. Co-training was first introduced in 1998, which was originally designed for independent redundant "multi-view" data and was a kind of semi-supervised learning method based on "divergence". The key to the algorithm is to assume that there are two independent redundant views. First, two classifiers with large differences are trained and the unlabeled sample set is classified. Subsequently, the positive and negative samples, respectively, with high confidence or that meet the set threshold are labeled, and then they complement each other by adding these samples to form a new view. Lastly, the filtered pseudo-label samples are deleted from the unlabeled sample set. Iterations occur, as described above, until the unlabeled sample set is empty or meets a specific stop criterion.

In Figure 1, the process of co-training is described. *View*1 and *View*2 denote the known view 1 and known view 2, respectively; *Test* denotes the test set or unlabeled sampling set.

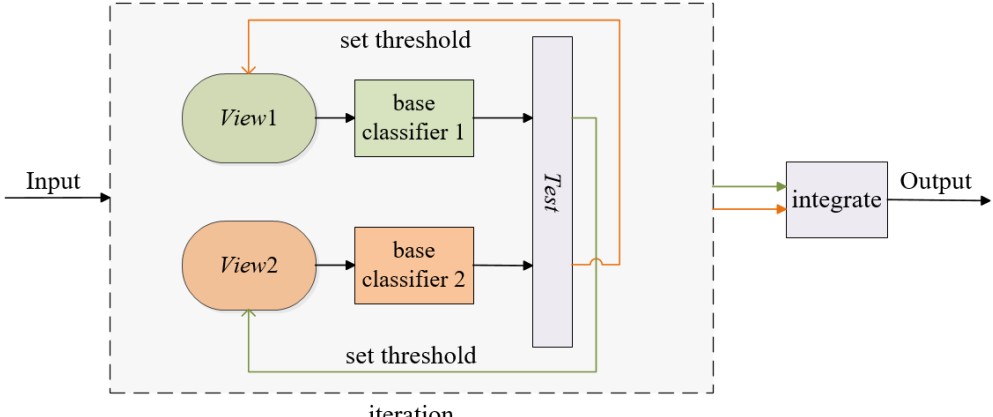

**Figure 1.** The process of co-training.

*3.2. Semi-Decoupled Features*

In this section, we introduce the research background of semi-decoupled features and the specific process of the algorithm. The feature set itself cannot be separated naturally. Therefore, it is difficult to ensure the independence between the views if it is randomly divided. That is, there is still a strong correlation between multiple views, which makes it difficult to realize the difference complementing of the independent views.

In the process of obtaining independent dual views based on random segmentation, we propose the innovation of semi-decoupled feature co-training. It mainly includes three steps: shared important features selection, remaining features separation, and features combination. First, the importance of the features is calculated and ranked [43,44], and a few important features are selected and added to the dual view together. In addition, we compute the correlation coefficients of the remaining features. Those with larger correlation coefficients are placed in the same view, while those with smaller correlation coefficients are divided into two different views so that the remaining features are retained or separated in two; then, the important features and separated features are combined. With the two views' acquisition based on semi-decoupled features, on the one hand, it ensures that the strong features continue to play an essential role in the semi-decoupled double view; on the other hand, it minimizes the interference between weak features and weak features in the process of model training and avoids the negative influence caused by attribute conflict between features as much as possible. The major calculations in this process are shown below.

3.2.1. Feature Importance

According to the *Gini* index and the random forest algorithm, feature importance is calculated and ranked. The smaller the *Gini* index, the better the feature attributes. The calculation formula is as follows, where $D$ denotes a certain dataset; $v$ and $V$ refer to the value of a feature and the total number of features respectively; $p_{f_x}$ means the proportion of positive cases of feature $f_x$ in the dataset; $p_{f_x}' = 1 - p_{f_x}$.

$$Gini(View) = \sum_{v=1}^{V} \sum_{f_x' \neq f_x} p_{f_x} \times p_{f_x}' = 1 - \sum_{v=1}^{V} p_{f_x}{}^2 \tag{1}$$

$$Gini(feature, f_x) = \sum_{v=1}^{V} \frac{|D^v|}{|D|} Gini(D^v) \tag{2}$$

The concept of shared features is displayed in Figure 2. The orange part of the *View* shows the important features, and the other two colored parts show the remaining features. The feature importance histogram was used to select the important features, and *View*1 and *View*2 both stand for the dual views after sharing the important features.

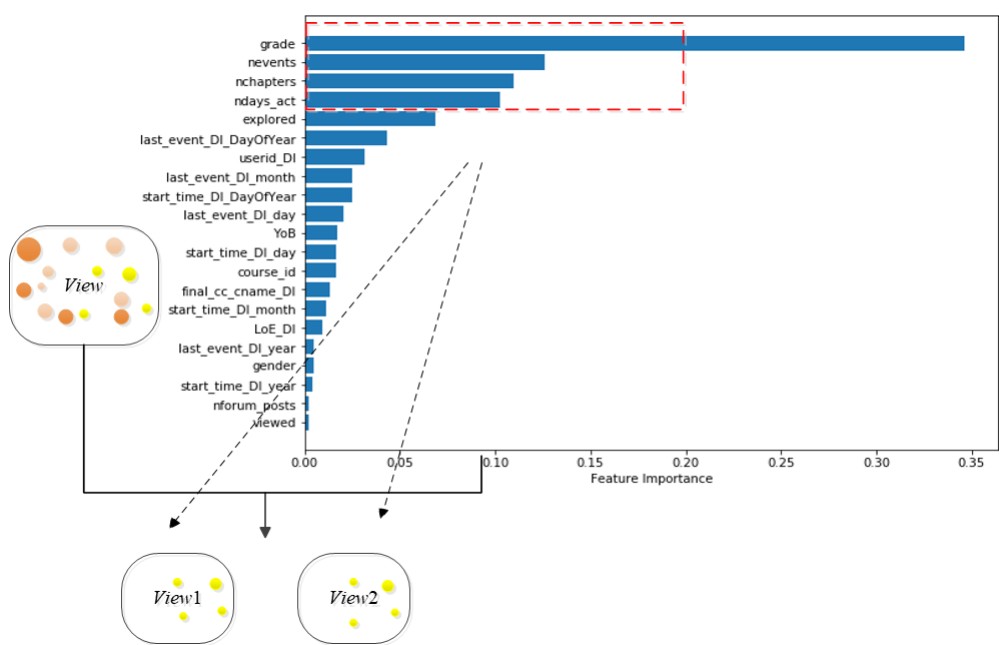

**Figure 2.** The selection of shared features. Orange parts: the important features, which are selected by feature importance; blue and green parts: the remaining features.

### 3.2.2. Correlation Coefficient between Features and Features

Through calculating the Pearson correlation coefficient and the assistance of a heat map, the remaining features are separated. The process needs to satisfy the following criteria:

- Those with weak feature correlation do not co-exist in the same view;
- Those with strong feature correlation must co-exist in the same view.

By following the above premise, the features are separated one by one or in small batches, and they are added to *View*1 or *View*2, accordingly, until the two views have completed the feature selection of the original single view. Then, the final redundant and more independent dual views, *View*1 and *View*2, are formed for co-training. The formula is shown below, where $f_x$ and $f_y$ stand for features $x$ and $y$; $\sigma_{fx}$ and $\sigma_{fx}$ represent the variance of features $x$ and $y$; $E(f_x)$ and $E(f_y)$ denote the mean values of features $x$ and $y$, respectively.

$$\rho_{f_x, f_y} = \frac{\mathrm{cov}\left(f_x, f_y\right)}{\sigma_{fx}\sigma_{fy}} = \frac{E\left[\left(fx - \mu_{fx}\right)\left(fy - \mu_{fy}\right)\right]}{\sigma_{fx}\sigma_{fy}} \tag{3}$$

The concept of remaining feature separation is presented in Figure 3. The blue part and the green part show the remaining features, and the orange part represents those important features. This figure extracts the correlation coefficients of some features, for example, to draw a heat map. *View*1 and *View*2 indicate that the single view *View* started to separate the remaining features after removing the shared features.

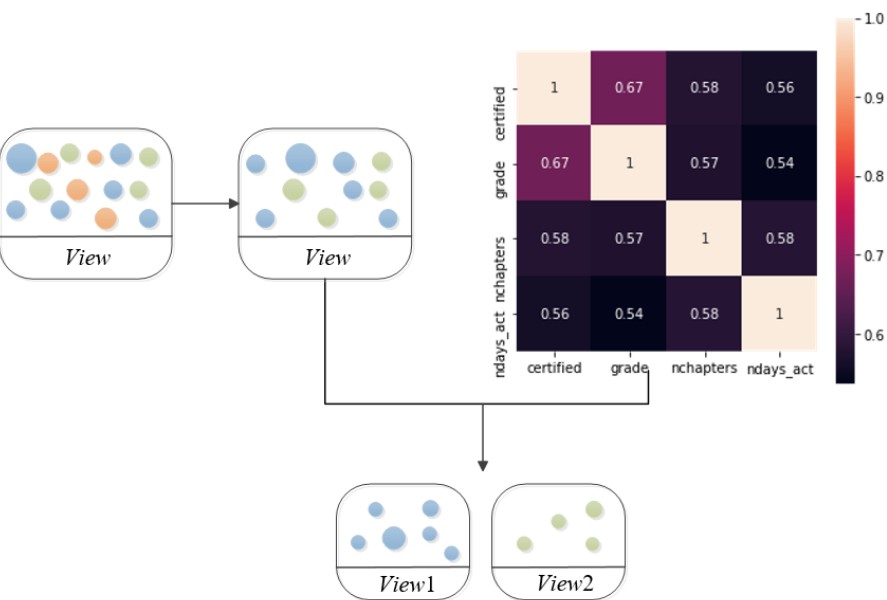

**Figure 3.** The separation of the remaining features. Orange parts: the important features; blue and green parts: the remaining features, which require to be separated.

### 3.3. The Algorithm of Semi-Decoupled Feature Co-Training

The algorithm is presented in Algorithm 1. Here, the samples of the single-view View are labeled, but the samples of Test are not. *View*1, *View*2 are empty; $\rho_{max}$, $\rho_{min}$ correspond to maximum threshold, minimum threshold; $w_1$ and $w_2$ mean the weights of the predictions from *View*1 and *View*2; the whole View includes n features, among them, the set of features $F = \{f_1, f_2, f_3, \ldots, f_y, \ldots, f_n\}$, where $f_y$ is the label of the sample.

---

**Algorithm 1:** Semi-decoupled feature co-training algorithm.

---

**Input:** Single-view *View*, *View*1, and *View*2; $\rho_{max}$ and $\rho_{min}$; $w_1$ and $w_2$; *Test*.
**Output:** *final_prediction*.
1: importance = *F*.feature_importances_().    // *F* = *F*.drop(*F*.$f_y$)
2: Sort importance.
3: Achieve *View*'s features sharing.
4: $\rho_{fx}$, $\rho_{fy}$ = *F*.corr($f_x$, $f_y$).    // *F* includes the feature of $f_y$
5: Divide remaining features.
6: Get *View*1, *View*2 with divergence.
7: **for** *iteration* = 1, 2, . . . **in** *iterations*:
8:       predict the *Test*.
9:       **if** *prediction* > $\rho_{max}$
10:           *prediction* = 1
11:       **if** *prediction* < $\rho_{min}$
12:           *prediction* = 0
13:       **end if**
14:        both classifiers no longer change or reach a predetermined
15:       number of iterative rounds.
16: **end for**

---

The process of the semi-decoupled feature co-training algorithm is shown in Figure 4. At first, the features are shared; then, those features represented by orange are separated in *View*1 and *View*2; in addition, the other features are separated by green into a *View*1 and the blue into a *View*2; then, the orange *View*1 and green *View*1 and the orange *View*2 and the blue *View*2 are combined to realize the construction of a new *View*1 and new *View*2; finally, the formal co-training can begin.

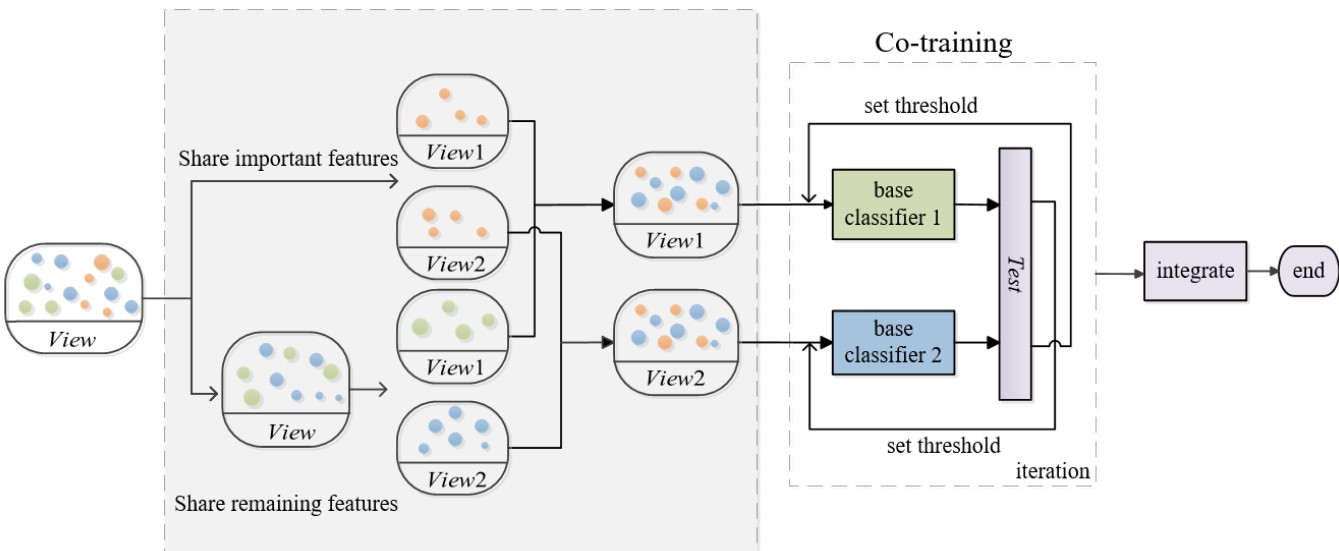

**Figure 4.** The process of semi-decoupled feature co-training. Orange parts: the important features; blue and green parts: the remaining features.

## 4. Results

### 4.1. Evaluation Metrics

The evaluation metrics used in the experiment included *F1*, *Precision*, and *Recall*.

- *F1*: The combination of *Precision* and *Recall*. *Precision* and *Recall* influence each other. If *Precision* increases then *Recall* decreases; if *Recall* increases then *Precision* decreases; if both need to be balanced, then *F1* measure is needed.

$$F1 = \frac{2}{\frac{1}{Precision} + \frac{1}{Recall}} \tag{4}$$

- *Precision*: The precision rate, which indicates the percent of the positive category samples that were actually positive. *TP* means that the original case was positive and was predicted to be positive; *FP* means that the original case was negative but was predicted to be positive.

$$Precision = \frac{TP}{TP + FP} \tag{5}$$

- *Recall*: The recall rate, which also refers to as the check-all rate, means the percentage of positive class samples marked as positive. *FN* represents cases that were originally positive but were predicted to be negative.

$$Recall = \frac{TP}{TP + FN} \tag{6}$$

### 4.2. Comparative Experiment

To verify the effectiveness of the innovation model and prevent overfittings in this paper, we introduced K-fold cross-validation to avoid a high variance and bias. First, we set the K-fold cross-validation fold number, *K*, as 2, 3, and 5. When *K* = 2, the proportion of the training set and test set was 5:5; when *K* = 3, the proportion of the training set and test set was 1:2 or 2:1; when *K* = 5, the proportion of the training set and test set was 2:8 or 8:2. Then, we compared our model with a traditional supervised model, three tree models, and three conventional semi-supervised models. The traditional supervised model used logistic

regression (LR) [45]; the integrated tree models used random forest (RF), LightGBM (LGB), and gradient boosting decision tree (GBDT) [46]; the conventional semi-supervised models used self-training [47], co-training, and tri-training. Among them, self-training used GBDT as the base model for screening pseudo-labeled samples; co-training used both LGB and GBDT as the base model to build classifiers with "divergence" [48]; tri-training used GBDT as the base model to vote so as to select the appropriate pseudo-labeled samples.

### 4.2.1. Test on the edX Dataset

To prove the effectiveness of the algorithm, we used the edX dataset [49], jointly published by Harvard University and MIT, as the sample set for the experiments. This release is composed of de-identified data from the first year (academic year 2013: fall 2012, spring 2013, and summer 2013) of HarvardX courses on the edX platform along with related documentation. There were 338,223 samples in total, and each sample contained 20 features. The specific features and their descriptions [50] are shown in Table 1. Among them, certified was the sample label and the criterion for the accuracy. The more similar the result was to the label, the more accurate the algorithm. When certified = 1, this meant that the certificate was obtained, and when certified = 0, this meant that the certificate was not obtained. The number of samples that obtained a certificate or not was 6570 and 331,653 respectively. The website for downloading and detailing data is https://doi.org/10.7910/DVN/26147 (accessed on 11 January 2022).

**Table 1.** The features table of the edX dataset.

| Feature | Feature Description | Feature | Feature Description |
|---------|--------------------|---------|--------------------|
| course_id | The id of all courses | grade | The grade of a course |
| userid_DI | The id of all users | start_time_DI | The start time of registration |
| registered | Whether to register for the course | last_event_DI | The last time of visit |
| viewed | Whether to access the courseware | nchapters | The learning chapter |
| explored | Whether to explore the process | ndays_act | The days of interaction |
| certified | Whether to obtain a certificate | nforum_posts | The number of forum posts |
| final_cc_cname_ | Nationality | nplay_video | The number of videos played |
| LoE_DI | Academic qualifications | nevents | The number of interactions in the course |
| YoB | Birthday | roles | The role that MOOC learner played |
| gender | Gender (male or female) | incomplete_flag | Whether the information is filled in completely |

Table 2 illustrates the *F*1, *Precision*, and *Recall* values of different models tested on the edX dataset with the different folds of K-fold cross-validation and different proportions. First, it can be seen that with the increase in the proportion of the training set, when *K* was the same, the score value for each model also increased, indicating that an increase in training samples can effectively improve the performance of the model. Second, the model metrics can roughly be divided into three intervals. The metrics of models LR and RF were basically below 0.9; the scores of models LGB, GBDT, self-training, co-training, and tri-training were all roughly below 0.99, which reflects a greater performance advantage than LR and RF; while the scores for our model in this paper were all about above 0.99, and the highest scores were achieved in almost every division proportion. These analyses fully illustrate the stability and superiority of the model in this paper. Furthermore, compared with the traditional co-training method, our model had significantly improved scores in each division proportion in which the *F*1 values increased by 2–3 percentage points, indicating the effectiveness of the semi-decoupled feature method proposed in this paper. In conclusion, this algorithm constructs multiple views with "moderate divergence" based on the perspective of semi-decoupled features, which not only ensures that the important features are shared to play their proper role but also reduces the negative impact caused by feature conflicts.

**Table 2.** Comparison of the results of various models on the edX dataset.

| K-Fold Cross-Validation | Division Proportion | Metric | LR | RF | LGB | GBDT | Self-Training | Co-Training | Tri-Training | Ours |
|---|---|---|---|---|---|---|---|---|---|---|
| K = 2 | 5:5 | F1 | 0.7345 | 0.8944 | 0.9273 | 0.9754 | 0.9669 | 0.9772 | 0.9781 | 0.9946 |
| | | Precision | 0.6562 | 0.8646 | 0.8995 | 0.9658 | 0.9519 | 0.9597 | 0.9712 | 0.9908 |
| | | Recall | 0.8339 | 0.9263 | 0.9568 | 0.9851 | 0.9823 | 0.9953 | 0.9851 | 0.9983 |
| K = 3 | 1:2 | F1 | 0.7351 | 0.8859 | 0.9314 | 0.9720 | 0.9686 | 0.9755 | 0.9712 | 0.9940 |
| | | Precision | 0.6581 | 0.8550 | 0.9046 | 0.9609 | 0.9574 | 0.9569 | 0.9585 | 0.9898 |
| | | Recall | 0.8326 | 0.9191 | 0.9600 | 0.9833 | 0.9801 | 0.9948 | 0.9843 | 0.9982 |
| | 2:1 | F1 | 0.7350 | 0.8927 | 0.9314 | 0.9769 | 0.9684 | 0.9777 | 0.9810 | 0.9941 |
| | | Precision | 0.6613 | 0.8708 | 0.9046 | 0.9675 | 0.9574 | 0.9594 | 0.9719 | 0.9893 |
| | | Recall | 0.8274 | 0.9158 | 0.9600 | 0.9865 | 0.9798 | 0.9966 | 0.9903 | 0.9990 |
| K = 5 | 2:8 | F1 | 0.7326 | 0.8753 | 0.9198 | 0.9674 | 0.9590 | 0.9727 | 0.9626 | 0.9935 |
| | | Precision | 0.6581 | 0.8497 | 0.8912 | 0.9553 | 0.9457 | 0.9536 | 0.9518 | 0.9902 |
| | | Recall | 0.8261 | 0.9025 | 0.9504 | 0.9799 | 0.9727 | 0.9927 | 0.9737 | 0.9968 |
| | 8:2 | F1 | 0.7350 | 0.8942 | 0.9322 | 0.9759 | 0.9685 | 0.9771 | 0.9832 | 0.9953 |
| | | Precision | 0.6603 | 0.8668 | 0.9028 | 0.9677 | 0.9538 | 0.9584 | 0.9754 | 0.9920 |
| | | Recall | 0.8288 | 0.9234 | 0.9636 | 0.9842 | 0.9837 | 0.9965 | 0.9910 | 0.9986 |

Figure 5 shows the confusion matrix for each model. It was tested on the edX dataset when $K = 2$ and train:test = 5:5. Figure 5a shows that the LR model obtained the maximum classification error with 0.0115. Figure 5f shows that ours obtained the minimum classification error with 0.0004, 0.0004 higher than the second minimum error produced by tri-training in Figure 5g.

As shown in Figure 6, the evaluation metrics and performance visualization of the different models were carried out. The semi-decoupled feature co-training algorithm presented in this paper all achieved the best results, followed by other semi-supervised models. The LR model showed the worst performance with metrics all lower than 0.85.

### 4.2.2. Test on the Breast Cancer Wisconsin Dataset

To prove the effectiveness of the model and test its robustness in depth, we also used the Breast Cancer Wisconsin Dataset [51] as the benchmark tested for testing. The dataset can be easily obtained through scikit-learn. There were 569 records in the dataset, with 357 benign and 212 malignant in total. Each sample contained 32 columns with 30 features, and the remaining two were Id and Diagnostic. Every sample was labeled with the benign and malignant results of diagnosis.

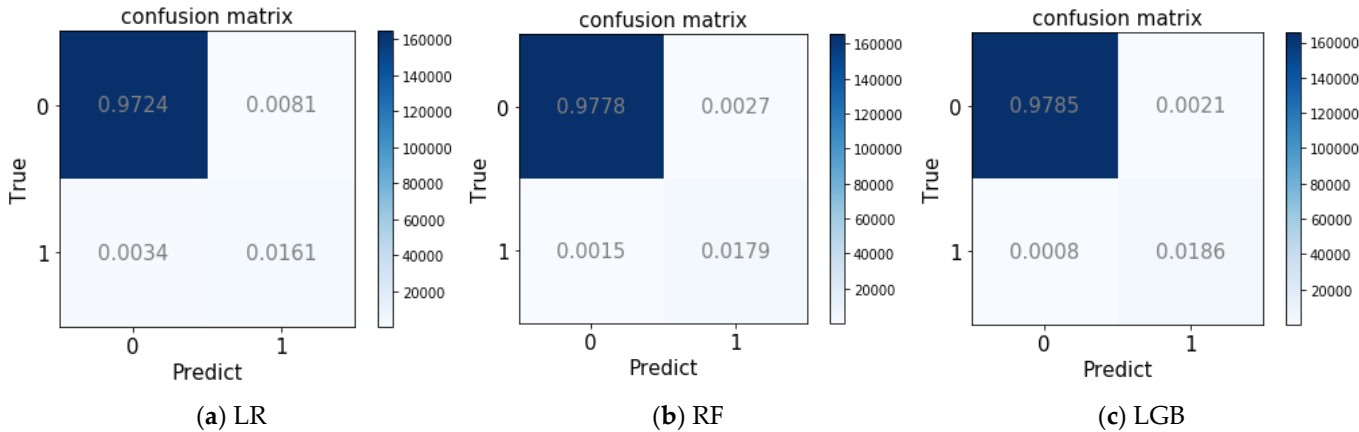

**Figure 5.** *Cont.*

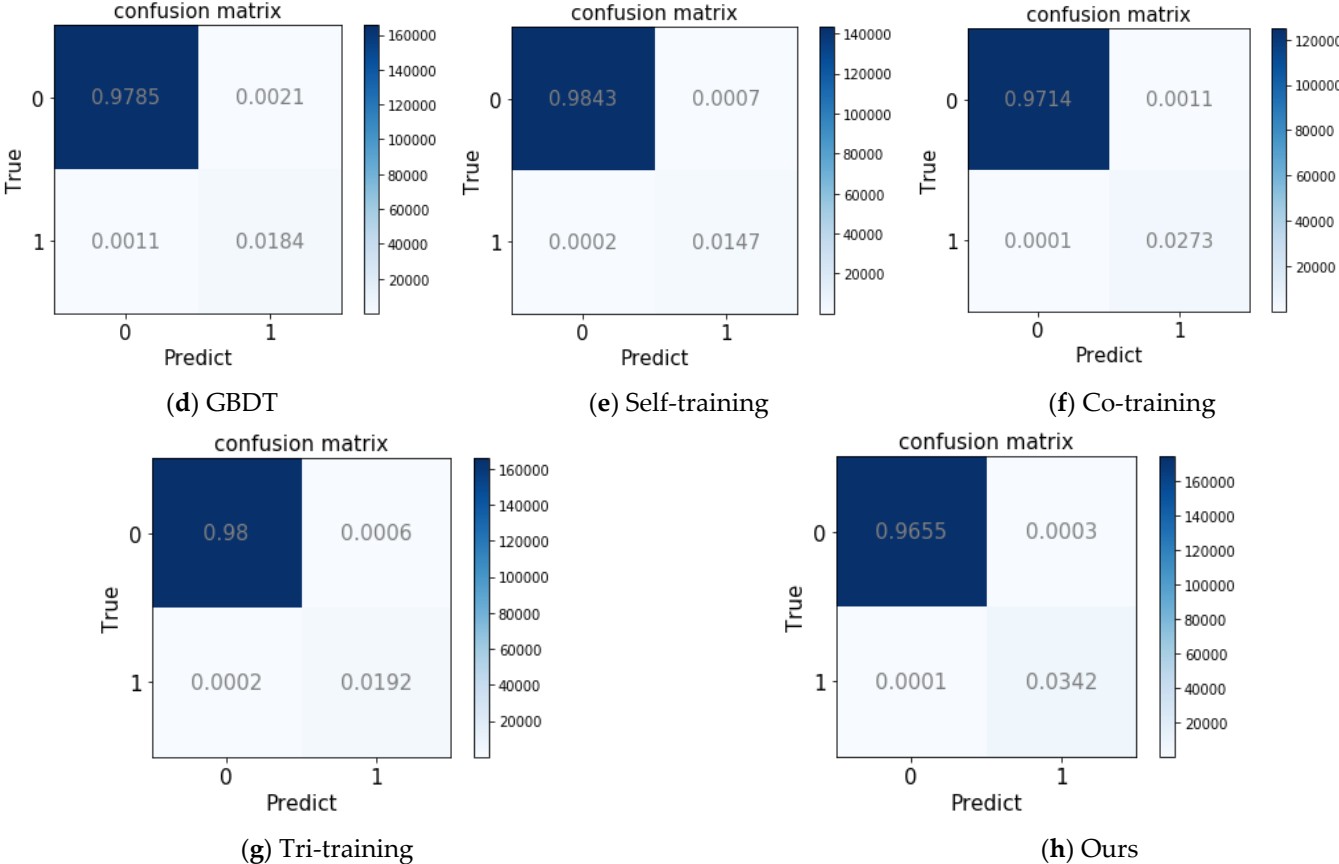

**Figure 5.** The confusion matrix for each model: (**a**–**h**) the confusion matrices obtained using the edX dataset with different models when $K = 2$ and train:test = 5:5.

Table 3 illustrates the *F1*, *Precision*, and *Recall* values of the different models tested on this dataset, using the different folds of K-fold cross-validation and different proportions. First, it can be seen that all models obtained good metrics of approximately 0.95 or more. Moreover, in the supervised model, RF presented the best performance; for the semi-supervised models, both self-training and co-training presented good metrics. However, in most cases, compared with the existing models stated above, our co-training model based on semi-decoupling features still retained the highest metrics when the $K$ and the proportion of the dataset were the same, which proves our model had the best performance once again.

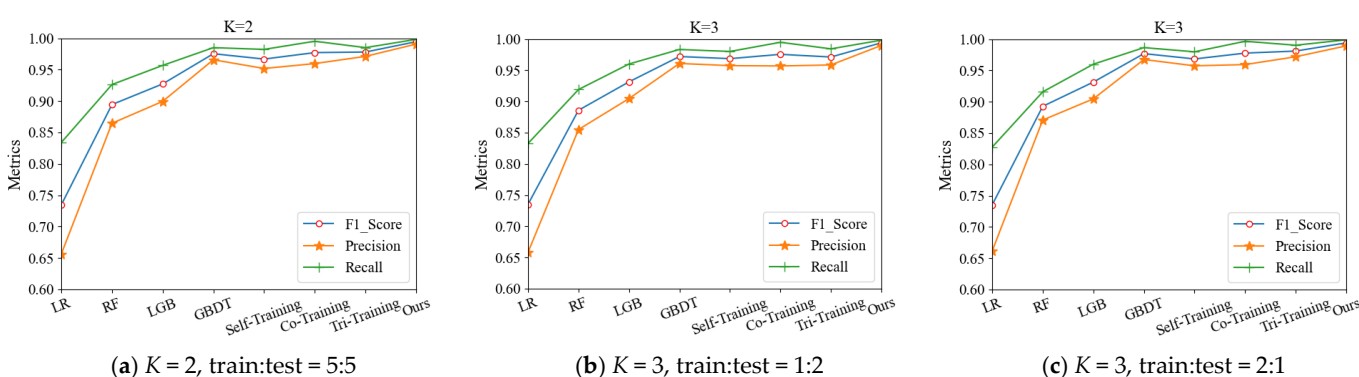

**Figure 6.** *Cont.*

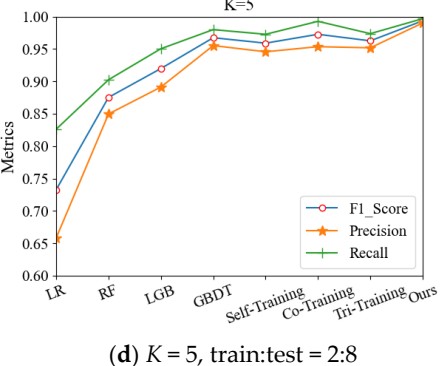
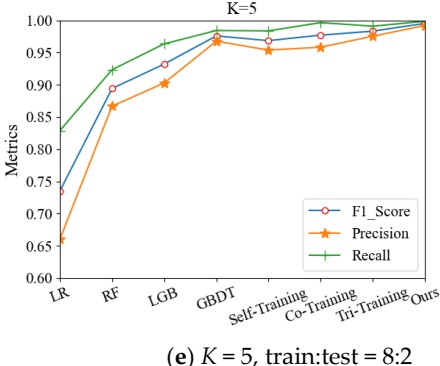

(**d**) *K* = 5, train:test = 2:8

(**e**) *K* = 5, train:test = 8:2

**Figure 6.** The evaluation metrics for each model: (**a**–**e**) the metrics obtained on the edX dataset with different models, which were tested under the different K-fold cross-validation and the different proportional divisions. Blue: *F1*; orange: *Precision*; green: *Recall*.

**Table 3.** Comparison of the results of various models on the Breast Cancer Wisconsin Dataset.

| K-Fold Cross-Validation | Train:Test | Metric | LR | RF | LGB | GBDT | Self-Training | Co-Training | Tri-Training | Ours |
|---|---|---|---|---|---|---|---|---|---|---|
| *K* = 2 | 5:5 | *F1* | 0.9541 | 0.9721 | 0.9720 | 0.9625 | 0.9760 | 0.9803 | 0.9791 | 0.9859 |
| | | *Precision* | 0.9506 | 0.9721 | 0.9724 | 0.9560 | 0.9674 | 0.9765 | 0.9777 | 0.9748 |
| | | *Recall* | 0.9581 | 0.9721 | 0.9721 | 0.9693 | 0.9847 | 0.9842 | 0.9804 | 0.9974 |
| *K* = 3 | 1:2 | *F1* | 0.9503 | 0.9678 | 0.9644 | 0.9477 | 0.9835 | 0.9831 | 0.9630 | 0.9872 |
| | | *Precision* | 0.9485 | 0.9665 | 0.9599 | 0.9313 | 0.9798 | 0.9755 | 0.9584 | 0.9747 |
| | | *Recall* | 0.9524 | 0.9692 | 0.9692 | 0.9650 | 0.9873 | 0.9908 | 0.9678 | 1.0000 |
| | 2:1 | *F1* | 0.9587 | 0.9723 | 0.9767 | 0.9555 | 0.9911 | 0.9909 | 0.9696 | 0.9860 |
| | | *Precision* | 0.9439 | 0.9617 | 0.9597 | 0.9247 | 0.9875 | 0.9847 | 0.9591 | 0.9725 |
| | | *Recall* | 0.9748 | 0.9832 | 0.9944 | 0.9888 | 0.9949 | 0.9974 | 0.9804 | 1.0000 |
| *K* = 5 | 2:8 | *F1* | 0.9432 | 0.9572 | 0.9536 | 0.9288 | 0.9755 | 0.9666 | 0.9578 | 0.9758 |
| | | *Precision* | 0.9344 | 0.9518 | 0.9498 | 0.9072 | 0.9660 | 0.9628 | 0.9531 | 0.9553 |
| | | *Recall* | 0.9524 | 0.9629 | 0.9580 | 0.9517 | 0.9860 | 0.9718 | 0.9629 | 0.9974 |
| | 8:2 | *F1* | 0.9574 | 0.9767 | 0.9794 | 0.9622 | 0.9899 | 0.9869 | 0.9767 | 0.9885 |
| | | *Precision* | 0.9491 | 0.9676 | 0.9729 | 0.9422 | 0.9900 | 0.9819 | 0.9678 | 0.9772 |
| | | *Recall* | 0.9667 | 0.9861 | 0.9861 | 0.9833 | 0.9899 | 0.9921 | 0.9861 | 1.0000 |

### 4.3. Iteration Analysis and Discussion

To evaluate the semi-decoupled feature model's performance at each iteration, on the edX dataset, we used K-fold cross-validation with *K* = 2, 3, and 5, and then we set the proportion of the training set and test set as 5:5, 1:2, 2:1, 2:8, and 8:2 as well. To explore the iteration's influence on the model, we conducted four experiments regarding iteration for analysis and discussion.

In Table 4, the results under different iterations of K-fold cross-validation and proportional division are shown. Top-down, it can be seen that the evaluation metrics increased unstably with the increase in the number of iterations and, overall, the best results were shown at iteration 2. Taking *K* = 2 and train:test = 5:5 as an example, for iteration 2, *F1*, *Precision*, and *Recall* were 0.9946, 0.9908, and 0.9983, respectively, which were 7–18 thousand points higher than for iteration 1. Then, the overall model improvement effect was gradually limited in iteration 3 and iteration 4. From left to right, when *K* was the same, with the increase in the samples in the training set, each metric of our model increased, and the highest metric of the model reached 0.9990. On the whole, Table 4 shows that the metrics improved and the performance was optimized with the number of iterations. However, with an increase in the pseudo-label samples, more noise samples will be introduced at the

same time. Therefore, the iteration should be stopped after iteration 2, which achieved the best effect in general.

**Table 4.** Analysis and discussion of the number of iterations.

| K-Fold Cross-Validation | Train:Test | Metric | Iteration 1 | Iteration 2 | Iteration 3 | Iteration 4 |
|---|---|---|---|---|---|---|
| K = 2 | 5:5 | *F1* | 0.9820 | 0.9946 | 0.9902 | 0.9891 |
| | | *Precision* | 0.9729 | 0.9908 | 0.9840 | 0.9815 |
| | | *Recall* | 0.9912 | 0.9983 | 0.9964 | 0.9969 |
| K = 3 | 1:2 | *F1* | 0.9804 | 0.9940 | 0.9892 | 0.9871 |
| | | *Precision* | 0.9710 | 0.9898 | 0.9827 | 0.9785 |
| | | *Recall* | 0.9900 | 0.9982 | 0.9957 | 0.9959 |
| | 2:1 | *F1* | 0.9825 | 0.9941 | 0.9906 | 0.9891 |
| | | *Precision* | 0.9735 | 0.9893 | 0.9845 | 0.9804 |
| | | *Recall* | 0.9917 | 0.9990 | 0.9967 | 0.9980 |
| K = 5 | 2:8 | *F1* | 0.9733 | 0.9935 | 0.9860 | 0.9857 |
| | | *Precision* | 0.9615 | 0.9902 | 0.9797 | 0.9791 |
| | | *Recall* | 0.9853 | 0.9968 | 0.9923 | 0.9925 |
| | 8:2 | *F1* | 0.9838 | 0.9953 | 0.9899 | 0.9895 |
| | | *Precision* | 0.9763 | 0.9920 | 0.9825 | 0.9806 |
| | | *Recall* | 0.9914 | 0.9986 | 0.9974 | 0.9986 |

Figure 7 visualizes the effect of our model at different iterations. In Figure 7, we divided the edX dataset into a training set and a testing set with K-fold cross-validation and proportion division. The best effect generated by the iterations was produced and stabilized in iteration 2, and all metrics declined gradually in iteration 3 and iteration 4. In iteration 2, the metrics increased slightly when K = 3, train:test = 2:1 and K = 5, train:test = 8:2, which was due to the greater number of labeled samples and less noise records.

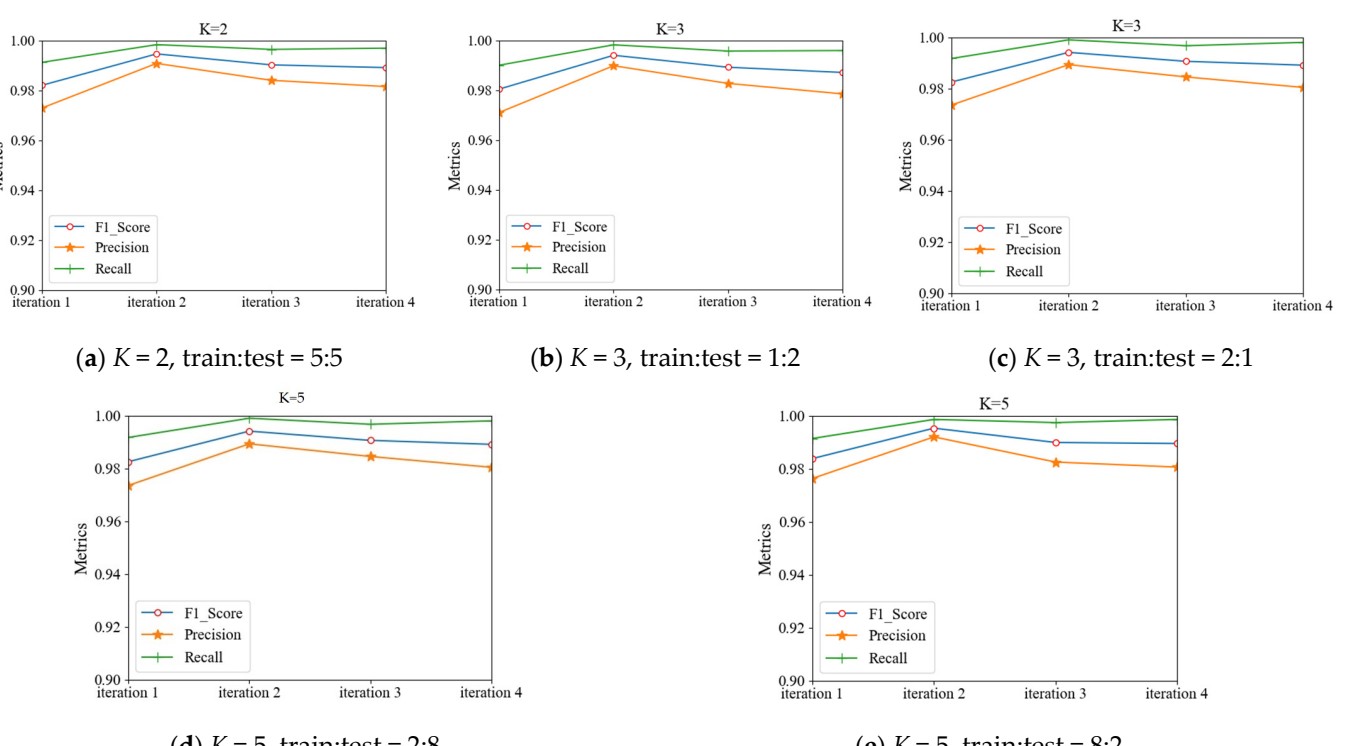

**Figure 7.** The iteration metrics of our model: (**a**–**e**) the metrics under the K-fold cross-validations and the percentage of divisions, which were tested on the edX dataset with four iterations. Blue: *F1*; orange: *Precision*; green: *Recall*.

*4.4. Hyperparameters*

In every experiment, the hyperparameters for the models' training were all important. Here, we introduce the hyperparameters of some important models tested in experiments: GBDT model, self-training model, co-training model, tri-training model, and co-training model based on semi-decoupling feature. We used GBDT with a learning rate = 0.1, n_estimators = 100, and random_state = 0. In the self-training model, co-training model, tri-training model, and the model proposed in this paper, we used GBDT as the base model and its parameters as the training hyperparameters. In addition, we added LGB as the other base model in co-training with n_estimators = 100, num_leaves = 12, colsample_bytree = 0.5, max_depth = 5, $\rho_{min} < 0.00001$, and $\rho_{max} > 0.5$. Among them, the hyperparameters $\rho_{min}$ and $\rho_{max}$ can be adjusted by the results of the models' prediction.

## 5. Discussion

In the comparative experiments of the multiple models, through *F*1, *Precision*, *Recall*, and the confusion matrix, our model showed the better performance and could improve the previous studies by semi-decoupled features. Compared with traditional supervised models, the co-training method based on a semi-decoupling feature model combined the unlabeled and labeled samples fully. Compared with the semi-decoupled models, such as co-training and tri-training, our model made the following main improvements: satisfied the assumptions of standard co-training multiple views; disentangled independent and redundant dual views as much as possible; avoided optimization of the difficulties and conflict problems for important feature attribute features.

The process of semi-decoupled features was mainly based on the importance of features and correlation between features so as to construct "divergent" views. On the one hand, we separated irrelevant features or redundant features based on the Pearson correlation coefficient and a heat map to reduce feature-to-feature interference; on the other hand, we enhanced the influence of important features on the model based on the calculation of feature importance by RF. Since the newly constructed dual views retained the relevant features, the model could achieve significant results. Meanwhile, those important features still continued to play an important role, and features with low correlation coefficients interfered less with each other after view segmentation.

In the process of semi-decoupling features, the importance of features can be calculated and selected not only by using the Gini index but also by combining the ANOVA f-test, mutual information, chi-square test, regression t-test, and variance check, etc., which may offer more reasonable and stable conclusions. The correlation between features can be assessed not only by the Pearson correlation coefficient but also by Spearman, Kendall, etc., in order to realize feature coexistence or separation.

This improved model is certainly applicable to semi-supervised learning of other tabular data, which can be achieved by semi-decoupling the data features into two tables with independence and redundancy. In addition, this work is not only limited to regular data but can be extended into the field of deep learning with non-tabular data.

Overall, this paper proposed a semi-decoupled feature co-training method that solves the current challenge, where the acquisition of multiple views is difficult to achieve. It improves the accuracy of the model, increases the high applicability of standard co-training, and opens up a further development prospect for semi-supervised learning.

## 6. Conclusions

In this paper, we proposed a co-training algorithm based on semi-decoupled features, which solves the problem of how to obtain fully redundant, conditionally independent dual views. It applied *Precision*, *Recall*, *F*1, and confusion matrix as the evaluation criteria to verify the generalization and learning performance of the training model with the algorithm. Through the experiments, we make the following conclusions.

Different from supervised learning and unsupervised learning, semi-supervised learning can learn on both the massive unlabeled data and limited labeled data at the same

time so as to improves the performance in depth. As a popular semi-supervised learning algorithm, co-training has become a research hotspot in the field of machine learning. Co-training requires sufficiently redundant and conditionally independent dual views, which is hard to achieve in real situations. Although many existing improvements generate different learners through different learning algorithms, different data adoptions, etc., they are still unable to meet the requirements of co-training. Thus, we proposed the co-training method based on a semi-decoupling feature algorithm. The innovate algorithm effectively solved the problem of independent view division, according to the two semi-decoupled standards of feature importance and correlation coefficients between features. This created both sufficient and different dual views for standard co-training. The experiments were conducted from three aspects: multiple models, iterations, and hyperparameters.

The contributions of the present study are three-fold. First, we innovated the algorithm in terms of view segmentation, which solved the common problem of massive single views but few multi-views in real scenes. Second, in co-training, there are some algorithms based on view segmentation that exist, but they still suffer from shortcomings with conflicting feature attributes and limitations of irrelevant features. In this study, we can avoid these disadvantages and can make more contributions: disentangling independent and redundant dual views on known single views as much as possible; satisfying the assumptions of standard co-training multiple views; avoiding the problems of conflicting feature attributes and difficulties in optimizing important features; improving the performance of the classifier. Third, the algorithm in this paper is really easy to achieve and effectively solves the problem that multiple views are difficult to see in many practical applications.

Although, the experiments prove that the algorithm in this paper has some improvements in co-training, there are still some limitations for future research to explore. First, with the increase in iterations, the noise samples added to the next training iteration will accumulate step by step and, consequently, its negative effect will become increasingly larger. Second, in a dataset with inadequate features, if the features are still semi-decoupled, the features used for training will be extremely scarce and may cause the results to be very poor. Third, in a dataset with enormous features, if we use feature semi-decoupling to realize feature separation according to feature importance and correlation coefficients, the workload will be very large.

As a summary of the above, the methodology used in this study can be extended to several solid future research directions. This paper did not discuss the problem of noise samples caused by pseudo-label samples. Thus, how to find and deal with these noisy data is a direction that deserves study in depth. In addition, in co-training, how to deal with those noise features, which are not important and have low Pearson correlation coefficients, is another direction. Furthermore, in a dataset with enormous features, with minimal workload, how to apply the idea of feature semi-decoupling is also a direction.

**Author Contributions:** Conceptualization, methodology, and formal analysis, H.W. and L.X.; writing—original draft preparation, experiment, and project administration, H.W.; validation and writing—review and editing, Z.H. and J.W.; resources, data curation, experiment, and supervision, H.W. and L.X. All authors have read and agreed to the published version of the manuscript.

**Funding:** This research was funded by the Science and Technology Innovation 2025 Major Special Project of Ningbo of China (grant number: 2019B10036) and the Natural Science Foundation of China (grant number: 62172356).

**Institutional Review Board Statement:** Not applicable.

**Informed Consent Statement:** Not applicable.

**Data Availability Statement:** Publicly available datasets were used in this study. EdX data can be downloaded at: https://doi.org/10.7910/DVN/26147, accessed on 11 January 2022.

**Conflicts of Interest:** The authors declare no conflict of interest.

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
