# Peer review of "Co-Training Method Based on Semi-Decoupling Features for MOOC Learner Behavior Prediction"

_axioms, doi:10.3390/axioms11050223_

Round 1
Reviewer 1 Report
First of all, the paper “Semi-supervised learning method based on semi-decoupling feature for MOOC learner behavior prediction” aims and scope match those of Axioms. However based on my opinion it needs substantial improvements to be considered for publication in Axioms. I would suggest a series of changes that in my opinion would improve the paper, in special for the reader.
- The abstract is loosely written. It is not as informative as expected. A standard abstract must present, without leaving any doubt, the objective of the paper precisely; source of data (which is not present in your abstract) and analytical approach used; key findings and any policy implication and recommendations.
- I suggest the authors to improve the introduction section. Authors should better highlight the objective of their work and to what extent it contributes to close a gap in the existing literature and/or practice. What is the innovative value of the contribution proposed by the authors?
- In introduction section authors should provide more information about existing training models and their benefits/weaknesses. The comparison with these methods should be presented. The authors need to discuss their contributions compared to those in related papers. The authors must clearly discuss the significance of the research problem in the first section.
- Why you have used semi-decoupled feature co-training algorithm in this study? Why not other algorithms?
- You should provide more recent references published in last two-three years in the Literature review. Remove references published before 2018. Discussing only 15 references in the Literature review is too low for scientific paper. You should extend the literature review section with more recent approaches in the uncertainty environment. I would like to suggest authors to read below interesting papers: Precup, R.-E., Preitl , S., Petriu, E., Bojan-Dragos , C.-A., Szedlak-Stinean, A.-I., Roman, R.-C., & Hedrea, E.-L. (2020). Model-Based Fuzzy Control Results for Networked Control Systems. Reports in Mechanical Engineering, 1(1), 10-25. https://doi.org/10.31181/rme200101010p; Ghosh, I., & Datta Chaudhuri, T. (2021). FEB-Stacking and FEB-DNN Models for Stock Trend Prediction: A Performance Analysis for Pre and Post Covid-19 Periods. Decision Making: Applications in Management and Engineering, 4(1), 51-84. https://doi.org/10.31181/dmame2104051g.
- Explain in more details in the data used in the case study, the data for the testing, the criterion for the accuracy, and others to claim these points.
- Validation section is not well prepared. How we can judge about these results? Comparisons with existing algorithms from the literature is missing.
- Discussion section is missing. How should we know about the quality of these solutions? Could you compare these results with some existing approaches in literature? The improvement must be discussed.
- The conclusion section seems to rush to the end. The authors will have to demonstrate the impact and insights of the research. The authors need to clearly provide several solid future research directions. Clearly state your unique research contributions in the conclusion section. Add limitations of the model. No bullets should be used in your conclusion section.
Reviewer 2 Report
The authors proposed a co-training algorithm based on semi-decoupled features, which solves the problem of how to obtain fully redundant, conditionally independent dual views and found that that the evaluation metrics of the semi-decoupled feature co-training model are significantly higher than those of other models. The manuscript has merits, but the following comments should be answered in the revised version of paper.
1. First, please mention and refer to the recent contributions on machine learning in Engineering with Computers, 2022, 1-26, 10.1007/s00366-021-01586-2, and Engineering with Computers, 2022, 1-22, 10.1007/s00366-022-01633-6.
2. Please specify the reason of using Gini index and random forest algorithm for feature importance measurement. There are other strategies for feature importance measurement, such as ANOVA f-test, Mutual Information, Chi-Square test, Regression t-Test and Variance Check, they may contribute to different rankings, however, a comprehensive measurement would offer more reasonable and stable conclusions.
3. For evaluation metrics, classification accuracy (CAR) also plays an important role, to show all those metrics, I would recommend using confusion matrix to graphically show the classification results and all those metrics.
4. The model evaluation was also not introduced. To prevent overfitting, a K-fold cross validation technique should be adopted to help to evaluate the model to avoid a high variance and bias in the results generated. The dataset is only divided into training and testing, the generality of the model is not tested upon unseen dataset, which makes the model less reliable. Besides, one sampling comparison is not convincing enough, u need to show the average results of Accuracy for a repeated sub-sampling process, that’s why repeated cross validation must be added here. Please add in the revision.
5. The hyperparameters for all machine learning models are important for readers to reestablish the model are not introduced in the manuscript, please add a hyperparameter table in the context. Also, please add introduction to the hyperparameter tuning and selection for the proposed model which will be important to prevent the model from getting stuck at local optimal and for a fair comparison.
6. The model is only tested upon one source of dataset (binary classification). To test the model robustness, it is advised to test the model on other benchmark datasets such as Iris plants dataset [1], wine recognition dataset [2], breast cancer wisconsin dataset [3] etc. Those datasets can be load easily through scikit-learn.
[1]. Fisher, R.A. “The use of multiple measurements in taxonomic problems” Annual Eugenics, 7, Part II, 179-188 (1936); also in “Contributions to Mathematical Statistics” (John Wiley, NY, 1950).
[2]. S. Aeberhard, D. Coomans and O. de Vel, Comparison of Classifiers in High Dimensional Settings, Tech. Rep. no. 92-02, (1992), Dept. of Computer Science and Dept. of Mathematics and Statistics, James Cook University of North Queensland. (Also submitted to Technometrics).
[3]. W.N. Street, W.H. Wolberg and O.L. Mangasarian. Nuclear feature extraction for breast tumor diagnosis. IS&T/SPIE 1993 International Symposium on Electronic Imaging: Science and Technology, volume 1905, pages 861-870, San Jose, CA, 1993.
Round 2
Reviewer 1 Report
I am very happy that the authors have addressed the point of my concern by point precisely. No further suggestions come from my side. Therefore, I would like to recommend this manuscript be published.
Reviewer 2 Report
The authors have modified their manuscript following all suggestions made by the reviewer. The reviewer support the publication of the manuscript.